# 10-Year Clinical Follow-Up after Decompression of Lipofibromatous Hamartoma of the Median Nerve in a 3-Year-Old Patient: Case Report and Review of the Literature

**DOI:** 10.3390/children10091581

**Published:** 2023-09-21

**Authors:** Seung Jin Yoo, Dae Hwan Kim, Seong Hyun Cho, Kyung Ryeol Lee, Kyu Bum Seo

**Affiliations:** 1Division of Hand and Microsurgery, Department of Orthopedic Surgery, Jeju National University Hospital, Jeju 63241, Republic of Korea; syoo06@gmail.com (S.J.Y.); dae1103@naver.com (D.H.K.); polehyun@naver.com (S.H.C.); 2Department of Radiology, Jeju National University Hospital, Jeju 63241, Republic of Korea; we1977@naver.com

**Keywords:** carpal tunnel release, lipofibromatous hamartoma, median nerve, pediatric carpal tunnel syndrome

## Abstract

Lipofibromatous hamartoma, first reported in 1953, is a rare, slowly progressive soft tissue tumor, the characteristics of which include the enlargement of the affected nerve via the epineurial and perineurial proliferation of adipose and fibrous tissues. Out of 200 previously reported cases of lipofibromatous hamartoma of the median nerve, there have been approximately 25 pediatric cases under the age of 18. Herein, we report a case of lipofibromatous hamatoma of the median nerve in a 3-year-old female patient who was surgically decompressed via carpal tunnel release and epineurolysis. The patient was followed-up on an outpatient clinic basis annually with sonographic evaluations, and the postoperative 10th-year follow-up did not show recurrence or any deficits in motor and sensory functions.

## 1. Introduction

Lipofibromatous hamartoma (LFH), first reported in 1953, is a rare, slowly progressive, benign soft tissue tumor characterized by the enlargement of the affected nerve by the epineurial and perineurial proliferation of adipose and fibrous tissues [1,2]. While most often found in the upper extremities, the median nerve is the most commonly reported nerve affected by LFH, but there have been previous studies reporting LFH occurring on other nerves, such as the ulnar, radial, and peroneal nerves [2]. Less commonly, the sciatic nerve and branches of the medial plantar nerve have also been reported to be associated with LFH [2]. The primary clinical manifestations are an enlarging mass and its nerve compression symptoms when it is large enough [3]. The most common (accounting for approximately 1/3 of cases) associated anomaly has been reported to be macrodactyly, a condition referred to as macrodystrophia lipomatosa [3]. However, the clinical presentations of LFH vary from asymptomatic to characteristic symptoms of compressive neuropathy [3].

While carpal tunnel syndrome (CTS) is the most common compressive neuropathy in the adult population, it is relatively rarely reported in pediatric populations [4]. While CTS is idiopathic in nature among adults, pediatric CTS is mostly secondary to anatomic variations, trauma, or congenital malformations [4]. LFH is the second-most common primary metabolic disorder causing CTS after mucopolysaccharidoses [2]. The two most discussed etiologies of LFH are repetitive microtrauma from the transverse carpal ligament to the median nerve and congenital anomalies [5]. Differential diagnoses vary from benign to malignant soft tissue tumors, such as ganglion cysts, neurofibromatosis, schwannoma, and malignant peripheral nerve sheath tumors [2].

Since the discovery of LFH in 1953, there have been approximately 200 cases of LFH of the median nerve and pediatric cases [6]. There have been about 25 cases of pediatric LFH afflicting those under the age of 18 [6,7,8,9,10,11,12,13,14,15,16,17,18,19,20,21,22,23,24,25,26,27,28,29,30,31,32,33]. However, even though most previous studies depended on case reports or small case series with short-term follow-up periods, some reported progressive decline in the sensory and motor functions of the affected nerve in long-term follow-ups of adult patients [8]. Herein, we report a case of a 3-year-old female patient with LFH on the left median nerve who was surgically decompressed via transverse carpal tunnel release with epineurolysis and was followed-up annually with ultrasonography assessments until the postoperative 10th year.

## 2. Case Description

A three-year-old female patient without underlying diseases visited the outpatient clinic at our institution with chief complaints of an enlarging mass and discomfort of the volar aspect of the left wrist. The patient’s mother recalled no history of trauma or congenital anomalies but reported a progressively enlarging solitary mass on the wrist for the past two weeks with mild pain in the beginning. The patient was otherwise unremarkable in terms of medical conditions and was not on any medications, and her immunizations were up to date.

Physical examination revealed a solitary, soft, mildly tender, palpable mass of approximately 1 × 1 cm on the center of the volar aspect of the wrist. There were no limitations in the ranges of motions of the wrist and fingers, but the patient’s agitation escalated with passive movement of the wrist joint. For this pediatric patient who still had difficulty with expressive language, it was difficult to subjectively assess Tinel’s sign, conduct Phalen’s test, or determine specific neuromuscular deficits. Gross inspection did not indicate any congenital abnormalities, such as polydactyly or café-au-lait spots.

Ultrasonographic examination on the patient’s first visit to the clinic showed a severely edematous and hypertrophied median nerve from the left-wrist level down to the palm with low-echogenic intraneural multiple fascicles (Figure 1). In addition, subsequent enhanced magnetic resonance imaging of the left wrist indicated fusiform enlargement of the median nerve from the left forearm distal 1/3 area down to the left hand metacarpal area along with characteristic findings of longitudinally oriented cable-like appearances of the tumor in addition to its invasion into the flexor retinaculum (Figure 2).

Upon the clinical and radiologic diagnosis of LFH, the patient was scheduled for a decompression of median nerve with a biopsy. A zig-zag incision on the volar aspect of the wrist was made from 10 cm proximal to the transverse carpal ligament down to the metacarpal joints area. Surgical exploration revealed a fusiform enlargement starting from 2 cm proximal to the transverse carpal ligament to the level of the thenar crease, surrounding the transverse carpal ligament and the median nerve (Figure 3). After the release of the transverse carpal ligament, longitudinal epineurolysis along the median nerve revealed abundant fibro-adipose tissue embedded in the nerve fascicles of the median nerve with severe adhesion. A hypertrophied epineurium was partially resected, and fibro-fatty tissue surrounding the epineurium was biopsied (Figure 3).

Histologic findings showed abundant thickened nerve bundles and accumulated fibrous and adipose tissues around the epineurium, confirming the diagnosis of lipofibromatous harmartoma in the median nerve (Figure 4).

Postoperatively, the patient’s recovery was uneventful, without any complications of pain, discomfort, or decreased joint movement. The patient was followed-up annually with sonographic examinations of the wrist (Figure 5). At the 10th-year follow-up, the patient remained asymptomatic, with full opposition function and grip strength without sensory deficits or further enlargement of the hamartoma in the median nerve, and electromyography and a nerve conduction study confirmed no neurologic deficits.

## 3. Discussion

Since its first report in 1953, LFH, a slowly growing soft tissue tumor, has been reported in 25 pediatric cases of median nerve involvement (Table 1). The previously reported incidence of hamartoma has shown a predilection for males, but a review of the previous studies, included in Table 1, on the occurrence of LFH indicated no predilection for race or country [34]. The countries of origin, in which LFH occurred, were widely distributed, encompassing Asia, America, and Europe. Except for pulmonary hamartoma, the epidemiology of hamartoma remains largely unknown [34]. The characteristic clinical manifestations of LFH involving median nerve are pain, paresthesia (i.e., numbness, tinging, and pins/needles), and motor deficits. Neurologic deficits, if present, are mostly irresponsive to conservative management and indicated for surgical management [2].

In the diagnosis of LFH, MRI plays a critical role by revealing pathognomic radiologic features and low-intensity serpentine nerve fibers embedded in abundant high-intensity adipose and fibrous tissues, also known as “cable-like appearances” [9]. In addition, ultrasonography also serves as an essential tool for initial diagnosis and postoperative non-invasive radiologic follow-ups. Ultrasound imaging of LFH is characterized by an enlarged cross-section of the affected nerve with hypoechoic fascicles embedded in the hyperechoic fibrous and adipose tissues, and recent ultrasonography has been shown to provide diagnostic evidence equivalent to that obtained using MRI [10]. While MRIs are beneficial in assessing the extent of a lesion in the initial diagnosis, ultrasonography serves a critical role in serial postoperative follow-ups, as seen in the current case description. However, biopsy and histologic examinations are the only definitive measures for the diagnosis of LFH, which are characteristic of intertwining collagen, fibroblasts, and adipose cells separating nerve fascicles and infiltrating the space between the epineurium and perineurium without inflammation or myelin degeneration [2].

Regarding the treatment of LFH, there is no standard treatment consensus, and patient management varies depending on the extent of the soft tissue lesion [5]. Historically, lipofibromatous lesions were completely excised surgically, with devastating sequelae of motor and sensory deficits [7]. However, the current treatment approach has been shifted toward a more conservative angle. Patients with asymptomatic swelling without severe neurologic deficits are frequently closely observed, but surgical treatments for patients with motor and sensory impairment aim to provide symptomatic relief from compressive neuropathy without invasive debulking [2,3,5]. There have been several previous studies on techniques for nerve dissections, but their postoperative functions are frequently disappointing [8,11]. In addition, nerve grafting after debulking and interaneural dissection have shown some positive results, but their results relied on short clinical follow-ups [12,13]. Consequently, tumor excision is generally reserved for cases with progressive and recurring pain and neurologic deficits even after carpal tunnel release and nerve decompression [14]. An interesting previous study on Martin-Gruber anastomosis, a neural anastomosis between the median and ulnar nerves, showed preserved hand and digital functions following the radical excision of the main trunk of the median nerve [15]. In addition, there was a case report in 2021 on the management of intractable postoperative chronic pain after the decompression of LFH of the median nerve via percutaneous peripheral nerve stimulation after the sonographic identification of the tumor boundaries, and the 1-year clinical follow-up data showed satisfactory pain reduction [35]. 

It is critical to consider a patient’s history, physical examination findings, and radiologic assessment data all together before making a decision on surgical treatment. As seen in the current case, when the patient is at the toddler age and incapable of expressive language, the clinical assessment of certain signs and symptoms is often limited and challenging, even in postoperative follow-ups. Furthermore, as it is a rare entity, it is difficult to compare various surgical techniques in terms of their effectiveness in treating LFH with a long-term follow-up. The currently available literature on pediatric LFH lacks long-term follow-up outcomes, and the corresponding cases are mostly followed-up for less than a few years. Therefore, the current case report provides valuable clinical and radiologic outcomes without tumor or symptom recurrence through annual follow-ups conducted consecutively for 10 years. Over the 10-year postoperative period, the patient was able to maintain intact motor and sensory functions without any pain or discomfort after the performed nerve decompression with the carpal tunnel release and epineurolysis of the median nerve.

## 4. Conclusions

Due to the rarity of LFH, a precise understanding of its epidemiology, treatment, and prognosis has been limited in the previous literature. The current case of LFH of the median nerve is the first to be reported in South Korea, and the high index of clinical suspicion for LFH is imperative, especially in pediatric carpal tunnel syndrome. Even though very little is known about the predictive prognostic values of surgical techniques in this regard, 10-year consecutive follow-ups of an initially 3-year-old patient with LFH showed satisfactory clinical outcomes after decompression with carpal tunnel release and epineurolysis. Furthermore, increased multidisciplinary clinical awareness of the tumorous condition is critical for best-practice management.

## Figures and Tables

**Figure 1 children-10-01581-f001:**
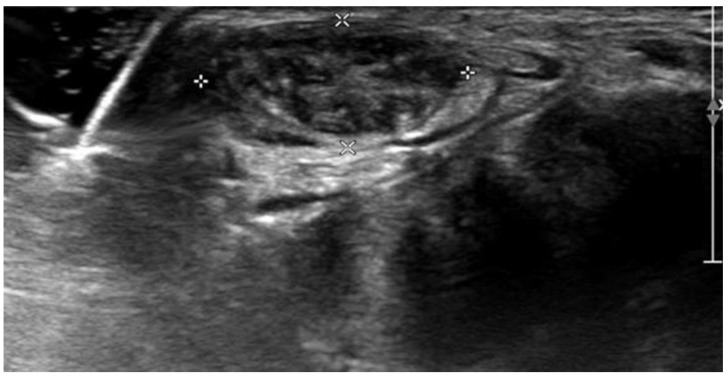
Wrist ultrasonography shows thickened and hypoechoic intraneural fascicles (white crosses).

**Figure 2 children-10-01581-f002:**
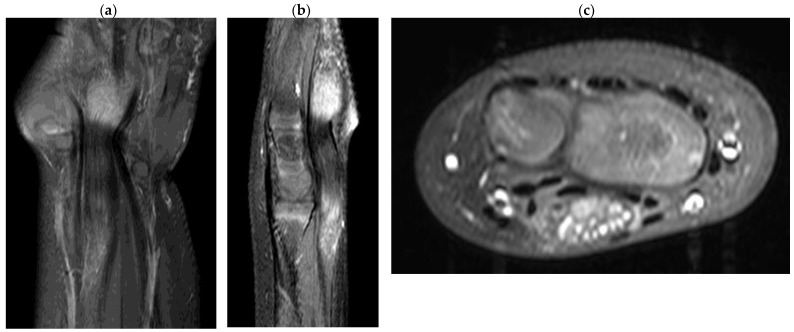
T1-magnetic resonance imaging of the lipofibromatous harmatoma in the median nerve. (**a**) coronal view; (**b**) sagittal view; (**c**) axial view.

**Figure 3 children-10-01581-f003:**
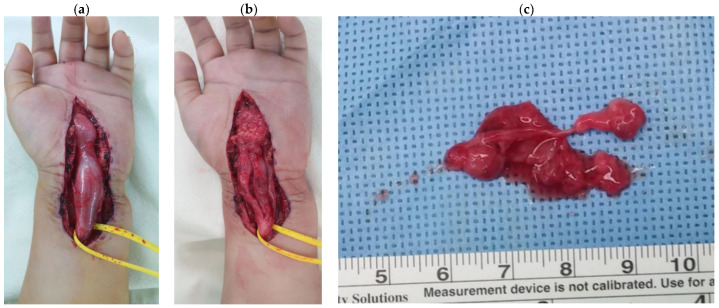
Intraoperative clinical images. (**a**) Enlarged median nerve with fibro-adipose tissue proliferation. (**b**) Decompressed median nerve with carpal tunnel release and epineurolysis and the partial excision of the mass for biopsy. (**c**) Biopsied soft tissues obtained during the epineurolysis.

**Figure 4 children-10-01581-f004:**
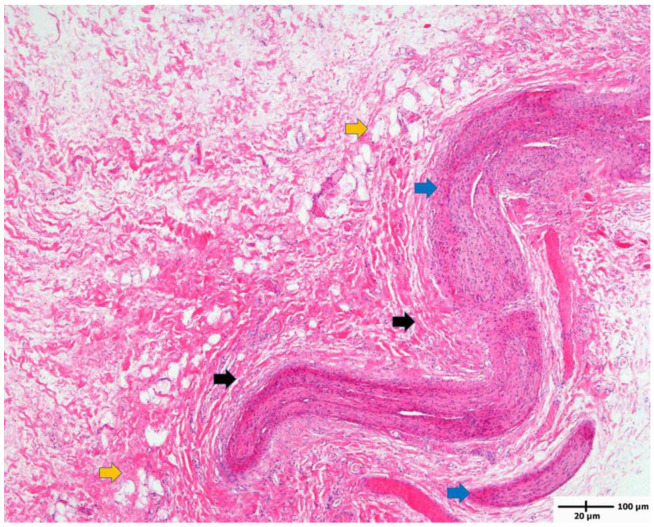
Characteristic histologic findings in lipofibromatous hamartoma (H&E stain, magnification ×40). The mass is composed of thickened nerve bundle (blue arrow) surrounded by accumulated fibrous tissue (collagen bundle, black arrow) and adipose tissue (yellow arrow).

**Figure 5 children-10-01581-f005:**
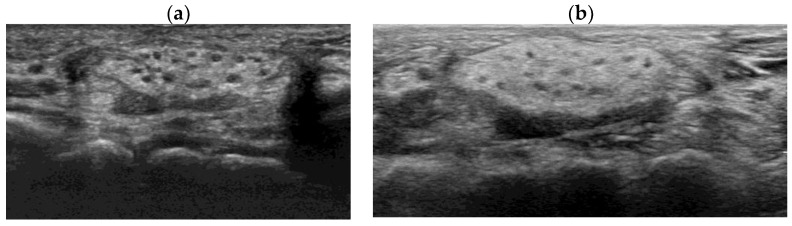
Ultrasonographic follow-up of left wrist at the postoperative (**a**) 5th and (**b**) 10th years.

**Table 1 children-10-01581-t001:** Pubmed/MedLine literature review of Lipofibromatous Hamartoma of the median nerve in the pediatric population (those aged under 18).

	Publication	Age (Year)	Gender	Laterality	Macrodactyly	Treatment	Follow-Up	Country
1	1987 [16]	4	M	L		Biopsy	-	Denmark
2	1987 [17]	14	F	L		CTR and partial debulking	-	USA
3	1991 [18]	3	M	L		CTR	-	USA
4	1998 [19]	3	M	L		CTR and excision	1Y	Netherland
5	2000 [20]	3	M	L		CTR and partial excision	-	USA
6	2006 [12]	16	M	R	3rd digit	CTR, debulking, nerve graft	4Y (Recur)	UK
7	2006 [12]	9	F	R		CTR and partial excision	10M	UK
8	2006 [21]	14	M	L		CTR, segmental excision, nerve graft	3Y	Taiwan
9	2008 [22]	5	F	R	3rd, 4th digits	Stripping of nerve		Belgium
10	2008 [23]	4	F	L		-	-	USA
11	2009 [24]	8	F	R	2nd digit	CTR and partial excision	8Y	Germany
12	2009 [25]	8	M	R		CTR and partial excision	-	India
13	2009 [26]	15	F	R		Observation	-	Belgium
14	2009 [26]	18	F	R		Observation	-	Belgium
15	2012 [27]	17	M	R		CTR and biopsy	6M	Japan
16	2012 [27]	15	M	R		CTR and biopsy	4Y	Japan
17	2014 [28]	3	M	Both		CTR and biopsy		Canada
18	2016 [6]	8	M	R		Observation	1Y	USA
19	2017 [13]	3	M	R		CTR, debulking, nerve graft	1Y	USA
20	2018 [29]	18	F	R		CTR	-	India
21	2018 [29]	16	F	R	Thumb	CTR and intraneural dissection	-	India
22	2018 [30]	10	F	R		CTR	2M	USA
23	2018 [31]	15	M	R		CTR and partial excision	3Y	Greece
24	2019 [32]	13	F	L		CTR	2Y9M	Iraq
25	2023 [33]	6	F	L		CTR and microsurgical interfascicular dissection	1Y	China
26	Case	3	F	L		CTR and biopsy	10Y3M	Republic of Korea

M: Male, F: Female, R: Right, L: Left, CTR: Carpal tunnel release, Y: Year(s), M: Month(s), USA: the United States of America, and UK: the United Kingdom.

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
