# Peer review of "10-Year Clinical Follow-Up after Decompression of Lipofibromatous Hamartoma of the Median Nerve in a 3-Year-Old Patient: Case Report and Review of the Literature"

_children, 2023, doi:10.3390/children10091581_

Round 1

Reviewer 1 Report

This case report is basically well written.
minor: I think Figure 1 is unnecessary.

Author Response

This case report is basically well written.
minor: I think Figure 1 is unnecessary.

; We removed the figure 1 as recommended.

Reviewer 2 Report

I would like to congratulate the authors on their fascinating work regarding this interesting case report on 10-year Clinical Follow-Up After Decompression of Lipofibromatous Hamartoma of the Median Nerve in a 3-year patient.

The manuscript is well-written and the incorporated table and figures make the study easy to follow.

1) I would like a brief discussion on Hamartoma (lung, breast, etc.) in general and its epidemiology.

Consider citing: https://www.ncbi.nlm.nih.gov/books/NBK562298/

2) Is there particular evidence of racial predilection for this disease?

Include in your table the "country" or "race" as additional information and make a brief discussion on that.

Author Response

) I would like a brief discussion on Hamartoma (lung, breast, etc.) in general and its epidemiology.

Consider citing: https://www.ncbi.nlm.nih.gov/books/NBK562298/

; We added a brief discussion on epidemiology of hamartoma with the referenced suggested.

2) Is there particular evidence of racial predilection for this disease?

Include in your table the "country" or "race" as additional information and make a brief discussion on that.

; We added the country column in the Table 1 so that readers and reviewers can identify each case with its origin of country. However, there have been no literature on racial predilection, and we added a brief discussion on no trends in the occurrence of lipofibromatous hamartoma associated with race or countries from the review of literature.

Reviewer 3 Report

Many thanks to the authors for having presented a so interesting case report about 10-year clinical follow-up after decompression of lipofibromatous hamartoma of the median nerve in a 3-year-old patient.”.

Title and Abstract

The title should be changed reporting that the authors performed also a review of the literature.

Key words

Please provide them in alphabetic order.

Background

·         Line 24: it should be explained if it is a malignant or benign tumor.

·         Lines 25-27: “Median nerve is the mostly commonly reported nerve of LFH, but there have been previous literatures of LFH occurring on other nerves, such as ulnar, radial, and peroneal nerves.” References should be provided.

·         There are several sentences without references.

·         Lines 34-35: the authors reported that CTS is idiopathic. This part should be improved mentioning also the role of fasciae in the pathogenesis of CTS (DOI: 10.1111/joa.13124).

·         Lines 36-39: this sentence should be rewritten.

·         At line 23and 109 it is reported that LFH was first reported in 1953, while at line 42 it reported in 1954. The authors should check and correct.

·         It should be reported what kind of treatment is suggested for this tumor. Surgery? What kind of surgery? Are the cases in which have been reported deficits treated surgically?

Methods

·         This section contains enough information to understand and possibly repeat the study.

·         You could divide in subgroups this section (i.e. surgical technique; follow up etc).

·         The captions of Figures should be placed below the figures as reported in the journal’s guidelines.

·         Letters of the figures should be placed under the figures.

·         The quality of Figure 3C should be improved.

·         Line 80: LFH abbreviation should be used.

·         Figure 5: scale bar should be added. 40X should be check. The quality of the figure should be improved.

·         It would have been useful/important to also evaluate the patient with questionnaires on hand functionality.

Discussion

·         The length and content of the discussion communicates the main information of the paper. The results presented have been discussed adequately with data provided in literature.

·         Lines 131-138: references are missing and should be added.

·         Since there are no long term follow up cases in literature you could add that future studies with longer follow ups could help reach a standard treatment of this pathology.

Conclusions

·         The conclusions reflect and refer to the results of the study. It is written in a schematic way, and it focuses the matters of the study.

Other comments

·         References should be placed before the full stop. The authors should check the Journal’s guidelines.

·         English needs to be improved.

Moderate editing of English language required

Author Response

Reviewer #3

Title and Abstract

The title should be changed reporting that the authors performed also a review of the literature.

; We changed the title as suggested

Key words

Please provide them in alphabetic order.

; We re-ordered the key words in an alphabetical order

Background

  • Line 24: it should be explained if it is a malignant or benign tumor.

; Added as suggested

  • Lines 25-27: “Median nerve is the mostly commonly reported nerve of LFH, but there have been previous literatures of LFH occurring on other nerves, such as ulnar, radial, and peroneal nerves.” References should be provided.

; Added as suggested

  • There are several sentences without references.

; Corrected as suggested

  • Lines 34-35: the authors reported that CTS is idiopathic. This part should be improved mentioning also the role of fasciae in the pathogenesis of CTS (DOI: 10.1111/joa.13124).
  • Lines 36-39: this sentence should be rewritten.

; Corrected as suggested

  • At line 23and 109 it is reported that LFH was first reported in 1953, while at line 42 it reported in 1954. The authors should check and correct.

; Simple typo was corrected.

  • It should be reported what kind of treatment is suggested for this tumor. Surgery? What kind of surgery? Are the cases in which have been reported deficits treated surgically?

; Even though I agree with the comment in part, the authors believe that without the standard treatment guideline, it lacks complete clinical evidence to suggest certain surgical techniques are beneficial or better than others when each case differs at the time of diagnosis in terms of size, extension, and symptoms. Even though current trends is shifting toward less invasive surgeries, we are reporting a long term result of decompression with epineurolysis with a review of literature. The current case is not just about the superiority of the current surgical technique used in the report. There have been a few cases with neurologic deficits, but these cases required more invasive surgeries, such as debulking or nerve graft, which seem a different topic than the current case.

Methods

  • This section contains enough information to understand and possibly repeat the study.
  • You could divide in subgroups this section (i.e. surgical technique; follow up etc).

; Even though I appreciate your recommendation, the current case seems not adequate to have surgical technique to be described since the surgical technique used in the current report does not differ from nor deal with anything novel from other similar LFH cases presented previously with simple decompression and epineurolysis.

  • The captions of Figures should be placed below the figures as reported in the journal’s guidelines.

; The captions of Figures were moved to below the figures.

  • Letters of the figures should be placed under the figures.

; Corrected as suggested

  • The quality of Figure 3C should be improved.

; Corrected as recommended. However, compared to the coronal and sagittal images, axial cuts seem to have poorer resolutions.

  • Line 80: LFH abbreviation should be used.

; Corrected as recommended

  • Figure 5: scale bar should be added. 40X should be check. The quality of the figure should be improved.

; Scale bar and 40x manifcation scale were added and confirmed, respectively.

  • It would have been useful/important to also evaluate the patient with questionnaires on hand functionality.

Discussion

  • The length and content of the discussion communicates the main information of the paper. The results presented have been discussed adequately with data provided in literature.
  • Lines 131-138: references are missing and should be added.

; Added as suggested

  • Since there are no long term follow up cases in literature you could add that future studies with longer follow ups could help reach a standard treatment of this pathology.

Conclusions

  • The conclusions reflect and refer to the results of the study. It is written in a schematic way, and it focuses the matters of the study.

Other comments

  • References should be placed before the full stop. The authors should check the Journal’s guidelines.

; Corrected as suggested

  • English needs to be improved.

; Revised manuscript was proofread by a native English speaker.